# High Exposure to Livestock Pathogens in Southern Pudu (*Pudu puda*) from Chile

**DOI:** 10.3390/ani14040526

**Published:** 2024-02-06

**Authors:** Ezequiel Hidalgo-Hermoso, Sebastián Verasay Caviedes, Jose Pizarro-Lucero, Javier Cabello, Rocio Vicencio, Sebastián Celis, Carolina Ortiz, Ignacio Kemec, Nour Abuhadba-Mediano, Ronie Asencio, Frank Vera, Carola Valencia, Rocio Lagos, Dario Moreira-Arce, Fernanda Salinas, Galia Ramirez-Toloza, Raul Muñoz-Quijano, Victor Neira, Rodrigo Salgado, Pedro Abalos, Barbara Parra, Simone Cárdenas-Cáceres, Nicolás A. Muena, Nicole D. Tischler, Itziar Del Pozo, Gorka Aduriz, Fernando Esperon, Sebastián Muñoz-Leal, Paula Aravena, Raúl Alegría-Morán, Raul Cuadrado-Matías, Francisco Ruiz-Fons

**Affiliations:** 1Fundacion Buin Zoo, Panamericana Sur Km 32, Buin 9500000, Chile; fernandasalinas505@gmail.com; 2Facultad de Ciencias Veterinarias y Pecuarias, Universidad de Chile, Av. Santa Rosa, Santiago 8820808, Chile; seba.verasay.caviedes@gmail.com (S.V.C.); jpizarro@uchile.cl (J.P.-L.); rocio.vicencio@ug.uchile.cl (R.V.); galiaram@uchile.cl (G.R.-T.); ra_muqui@hotmail.com (R.M.-Q.); victorneira@u.uchile.cl (V.N.); rodrigo.salgadomoya@gmail.com (R.S.); pabalos@uchile.cl (P.A.); fafalita.vet@gmail.com (B.P.); 3Laboratorio Clínico, Hospital Veterinario SOS Buin Zoo, Panamericana Sur Km 32, Buin 9500000, Chile; rocio.lagos.mv@gmail.com; 4Centro de Conservación de la Biodiversidad Chiloé-Silvestre, Nal Bajo, Ancud 5710000, Chile; javiercabellostom@chiloesilvestre.cl (J.C.); asenciorojel@gmail.com (R.A.); 5Departamento de Veterinaria, Parque Zoológico Buin Zoo, Panamericana Sur Km 32, Buin 9500000, Chile; scelis@buinzoo.cl (S.C.); cortiz@buinzoo.cl (C.O.); registrador@buinzoo.cl (I.K.); 6Escuela de Medicina Veterinaria, Universidad Mayor, Camino La Pirámide 5750, Santiago 7580506, Chile; nour.abuhadba@mayor.cl; 7School of Veterinary Medicine, Facultad de Ciencias de la Naturaleza, Universidad San Sebastian, Patagonia Campus, Puerto Montt 5480000, Chile; frank.vera@uss.cl (F.V.); cvalenciasoto@gmail.com (C.V.); 8Departamento de Gestión Agraria, Universidad de Santiago de Chile (USACH), Santiago 9170022, Chile; moreira.dario@gmail.com; 9Institute of Ecology and Biodiversity (IEB), Santiago 7750000, Chile; 10Escuela de Geografia, Universidad de Chile, Santiago 8820808, Chile; 11Laboratorio de Virología Molecular, Fundación Ciencia & Vida, Av. del Valle Nte. 725, Huechuraba, Santiago 8580704, Chile; scardenas@cienciavida.org (S.C.-C.); nmuena@cienciavida.org (N.A.M.); ntischler@cienciavida.org (N.D.T.); 12Facultad de Medicina y Ciencia, Universidad San Sebastián, Providencia, Santiago 8420524, Chile; 13Department of Animal Health, NEIKER-Basque Institute for Agricultural Research and Development, Basque Research and Technology Alliance (BRTA), Parque Científico y Tecnológico de Bizkaia, P812, 48160 Derio, Spain; ipozo@neiker.eus (I.D.P.); gaduriz@neiker.eus (G.A.); 14Veterinary Department, School of Biomedical and Health Sciences, Universidad Europea de Madrid, C/Tajo s/n, 28670 Villaviciosa de Odón, Spain; fernando.esperon@universidadeuropea.es; 15Departamento de Ciencia Animal, Facultad de Ciencias Veterinarias, Universidad de Concepción, Chillán 3812120, Chile; seba.munozleal@gmail.com (S.M.-L.); paularavena@udec.cl (P.A.); 16Escuela de Medicina Veterinaria, Sede Santiago, Facultad de Recursos Naturales y Medicina Veterinaria, Universidad Santo Tomás, Ejercito Libertador 146, Santiago 8370003, Chile; ralegria2@santotomas.cl; 17Health & Biotechnology (SaBio) Group, Instituto de Investigación en Recursos Cinegéticos IREC (CSIC-UCLM-JCCM), 13005 Ciudad Real, Spain; raul.cuadrado@uclm.es (R.C.-M.);; 18CIBERINFEC, ISCIII—CIBER de Enfermedades Infecciosas, Instituto de Salud Carlos III, 28029 Madrid, Spain

**Keywords:** *Leptospira interrogans*, *Pestivirus*, ELISA, *Chamydia abortus*, conservation, pudu, serosurveillance

## Abstract

**Simple Summary:**

Livestock diseases can affect the health of wild ruminants, and some of them are zoonotic, affecting the human health, and additionally, wildlife can act as excellent sentinels for infectious disease, since they have limited home ranges. To gain a better understanding of the disease epidemiology of livestock and zoonotic pathogens, we examined the prevalence of antibodies against *Brucella abortus*, *Chlamydia abortus*, *Coxiella burnetii*, seven pathogenic serovars of *Leptospira interrogans* (Bratislava, Ballun, Grippotyphosa, Pomona, Canicola, Hardjo and Coppehageni), *Mycobacterium bovis*, *Toxoplasma gondii*, *Neospora caninum*, SARS-CoV-2, Hepatitis E Virus, *Pestivirus*, Bovine Herpesvirus-1 (BHV-1), Epizootic Hemorrhagic Disease Virus (EHDV), and Bluetongue Virus in 164 wild and under-human-care pudus from central and southern Chile using several serological tests. We detected high seroprevalences for *Leptospira interrogans* Harjo and *Pestivirus* in wild pudus, suggesting a livestock transmission in the template forest, and for *T. gondii* in under-human-care animals. A *Pestivirus* outbreak is the most strongly suspected as the cause of abortions in a zoo in the past. This study presents the first evidence of *Chlamydia abortus* in wildlife in South America and exposure to *Toxoplasma gondii*, *Leptospira interrogans*, and *Neopora caninum* in wild ungulate species in Chile, and further research will be necessary to understand their impact in the health and conservation of pudu.

**Abstract:**

A significant gap in exposure data for most livestock and zoonotic pathogens is common for several Latin America deer species. This study examined the seroprevalence against 13 pathogens in 164 wild and captive southern pudu from Chile between 2011 and 2023. Livestock and zoonotic pathogen antibodies were detected in 22 of 109 wild pudus (20.18%; 95% CI: 13.34–29.18) and 17 of 55 captive pudus (30.91%; 95% CI: 19.52–44.96), including five *Leptospira interrogans* serovars (15.38% and 10.71%), *Toxoplasma gondii* (8.57% and 37.50%), *Chlamydia abortus* (3.03% and 12.82%), *Neospora caninum* (0.00% and 9.52%), and *Pestivirus* (8.00% and 6.67%). Risk factors were detected for *Leptospira* spp., showing that fawn pudu have statistically significantly higher risk of positivity than adults. In the case of *T. gondii*, pudu living in “free-range” have a lower risk of being positive for this parasite. In under-human-care pudu, a *Pestivirus* outbreak is the most strongly suspected as the cause of abortions in a zoo in the past. This study presents the first evidence of *Chlamydia abortus* in wildlife in South America and exposure to *T. gondii*, *L. interrogans*, and *N. caninum* in wild ungulate species in Chile. High seroprevalence of livestock pathogens such as *Pestivirus* and *Leptospira* Hardjo in wild animals suggests a livestock transmission in Chilean template forest.

## 1. Introduction

The pudu (*Pudu puda*) is one of the smallest cervid species in the world. It is native to the temperate dense scrub forests of Chile (36–49° S) and Argentina (39–43° S) [1]. With a population in Chile of between 5000 and 10,000 animals, it is the most common cervid in the southern cone. Its breeding and reproduction in captivity being common practices, an estimated additional 300 animals are under human care in Chile [2]. However, its wild population has been declining in recent years due to anthropogenic causes and landscape changes [3], and it is currently classified as a vulnerable species according to Supreme Decree No. 151 [4].

Among the issues that could be threatening this species are livestock diseases [5,6], which have caused declines in the number of individuals in wild ruminant populations in North America, Asia, Africa, and Europe over the last quarter of a century [7,8,9,10,11,12]. These events could be due to the increasing interaction rates of livestock and wildlife because of the increase in the population globally [13,14]. Latin America and the Caribbean have also presented in recent decades an increase in livestock population and animal production [15]. However, currently there are no wild ruminant monitoring programs for livestock diseases, which results in knowledge gaps with respect to disease occurrence, identification of reservoirs and their role, and the infectious dynamic of these diseases [16]. Assessing pathogen exposure is critical, as reservoir hosts for novel pathogens are often identified from the results of serological assays. This happens even before the isolation of the pathogen itself, so it can be very useful [17], especially in regions with little epidemiological information, such as Latin America and the Caribbean and countries like Chile, where knowledge about the epidemiology of livestock and zoonotic diseases in wild ruminants is still limited.

Serological studies suggest that wild cervids could be susceptible and represent a wildlife reservoir for zoonotic pathogens, such as SARS-CoV-2 in white-tailed deer (*Odocoileus virginianus*) in the USA or bovine tuberculosis in red deer (*Cervus elaphus*) in Spain [18,19,20]. This justifies their health monitoring as sentinels for relevant diseases in wild environments [21,22,23]. In Chile, Salgado et al. [5] provided evidence of pudu abortions in zoos whose pathology, including congenital malformations, suggests bovine viral diarrhea as the cause. However, the causal agent could not be confirmed. Recently, two new intracellular bacteria, *Mycoplasma ovis*-like, *Anaplasma phagocytophillum*, and a new intracellular protozoan, *Babesia* sp., were identified in Chilean wild pudus, which could have zoonotic potential with public health implications [24,25,26]. The objective of this study was to estimate the existence of exposure and the risk factors influencing it in relation to *Brucella abortus*, *Chlamydia abortus*, *Coxiella burnetii*, *Leptospira interrogans* serovars (Bratislava, Ballun, Grippotyphosa, Pomona, Canicola, Hardjo and Coppehageni), *Mycobacterium bovis*, *Pestivirus*, Bovine Herpesvirus-1, Bluetongue Virus, Epizootic Hemorrhagic Disease Virus, Hepatitis E Virus, SARS-CoV-2, *Toxoplasma gondii*, and *Neospora caninum*, in captive and free-ranging pudus from Chile from 2011 to 2023.

## 2. Materials and Methods

### 2.1. Serum Samples

Pudu sera were obtained from the serum banks maintained by three Chilean rehabilitation centers (Chiloe Silvestre, Chiloe Island, Los Lagos District; Universidad San Sebastian Veterinary Faculty, Puerto Mont, Los Lagos District; Universidad de Concepcion Veterinary Faculty, Chillan, Ñuble District), and two under-human-care populations (Buin Zoo, Buin, Metropolitan District; Fundacion Romahue, Los Lagos District) (Figure 1). All the animals come originally from all over its known distribution range in Southern Chile, from Maule district to Chiloe Island, and were collected between 2011 and 2023 by the veterinary staff of these centers. Sources for these serum banks included animals subjected annually to medical check-ups in under-human-care centers. In the rehabilitation centers, most of the animals enter because of health issues, mainly dog attacks, infectious diseases, vehicle collisions and other causes. Sera were stored in individual cryo-vials and frozen at −20 °C until use for serology. For this study, we used samples from all the animals available at the serum banks. Epidemiological information for wild and captive pudus, including sample origin, sampling date, health status, sex, and age, was gathered from each animal, whenever possible.

Although blood samples from 109 free-ranging (S1) and 55 captive (S2 and S3) pudu were collected, due to insufficient serum volume from some individuals, the sample size for specific serologic tests varied between 17 and 145 individuals, depending on the test. For all the free-ranging pudus, only one sample by pathogen was analyzed; however, for captive pudus, some individuals had one sample by pathogen (S2), and several others were monitored longitudinally (S3).

### 2.2. Laboratory Analyses

Commercially available tests were performed according to the manufacturer’s instructions (Table 1) to demonstrate active or former infection. The entire test has been used in cervid species serosurveys.

### 2.3. Data Analysis

All results are expressed as relative and absolute frequencies to determine the seroprevalence/positivity rate for each of the studied pathogens.

Statistical differences in positivity to each studied pathogen were estimated. The analysis was performed by calculating seroprevalence differences and 95% confidence intervals for the differences (when possible), based on a chi-square approach and testing that positivity rate differences were equal to 0 [44,45]. Statistically significant differences were set at *p*-value < 0.05.

A logistic multivariable regression was performed to determine risk factors associated with the positivity of each studied pathogen [46], considering that the outcome is negative (0) or positive (1) for each studied pathogen. Sex, age, and condition (captive or free-ranging) of the individuals were recorded. Models were built using a stepwise backward elimination process, the Likelihood Ratio Test (LRT) was used for model selection [47], and variables were kept in the model when the LRT gave a significant effect on the removal (*p* < 0.05) or if the estimations changed over 20% when removed, as this is an indicator of a potential confounding effect [46]. The convergence of the models was set to a value of epsilon (ε) = 𝑒^−16^ to guarantee an adequate level of stringency for the models performed. The Hosmer–Lemeshow test was performed for model adjustment evaluation. McFadden pseudo-R^2^ was also estimated, to have an estimation of the quality of the prediction of the outcome [48]. Sera with unknown sex or age were not included in the analysis. Biological and epidemiological coherent interaction between recorded factors was performed.

All the analyses were performed using R version 4.2.2 [49], and “fmsb” [50], “nlme” [51], “lme4” [52], “car” [53], “ggplot2” [54] packages for the multivariable logistic regression, and “ResourceSelection” [55] package for the seroprevalence differences estimations.

## 3. Results

### 3.1. Overall Exposure

Overall antibody prevalence against the livestock and zoonotic pathogens in pudu was 20.18% (22/109; 95% CI: 13.34–29.18) in free-ranging and 30.91% (17/55; 95% CI: 19.52–44.96) in captive. Free-ranging pudus presented antibodies against four pathogens; the highest seropositivity rate was found for *L. interrogans* and the lowest for *C. abortus* (Table 2). Under-human-care pudus presented antibodies against five pathogens, the highest seropositivity rate was found for *Toxoplasma gondii* and the lowest for *Pestivirus* (Table 2). Reactivity to *Pestivirus* was confirmed by VNT in five positive sera.

Of captive pudus sampled longitudinally (S3), some became seropositive during the study, indicating recent exposure, for *T. gondii*, *L. interrogans*y *Ch. abortus* inside the facility. The coinfection was 12.72% in under-human-care pudus but with no occurrence in free-ranging animals.

### 3.2. Leptospira interrogans Serovars

Four serovars were detected in wild (Hardjo—9.2%, Pomona—1.5%, Gryppotiphosa—3.0%, and Coppenhageni—3.0%) and two in captive (Gryppotiphosa—7.1%, and Hardjo—3.5%) pudus with titers of 1:100, except one free-ranging pudu with titers of 1:200 to serovar Pomona. Coinfection with different leptospiral serovars occurred only in one (2.8%) wild pudu (Hardjo and Gryppo serovars).

### 3.3. Data Analysis

Positivity rate differences were estimated for those pathogens/diseases with at least one positive individual (Table 3). It can be observed that statistically significant differences between under-human-care and free-range individuals were detected only for *Toxoplasma gondii*, indicating higher values among under-human-care individuals (Table 2).

Multivariable logistic regressions only detected significant associations between the tested factors and seropositivity with *Leptospira* spp. and *Toxoplasma gondii* models (Table 3). The *Leptospira* spp. model indicates that pudu fawns have a higher risk of being positive than adults. The *Toxoplasma gondii* model shows that pudu living in “free-range” have a lower risk of being positive to *T. gondii* (Table 3). None of the evaluated interactions resulted in any statistically significant result. The Hosmer–Lemershow test indicates a good adjustment between data and the models (*Leptospira* spp. *p* = 0.234; *Toxoplasma gondii* = 0.447). McFadden pseudo-R^2^ was estimated for both models, resulting in 12.61% for the *Leptospira* spp. Model, and 19.16% for the *Toxoplasma gondii* model.

## 4. Discussion

Surveillance for zoonotic and livestock pathogens in wild animal species is a critical step to our understanding of the epidemiology and control of infectious diseases at the interface between humans, domestic animals, and wildlife [56]. The use of sentinel wildlife species, like pudu, can be a useful tool for public health, livestock production, and prevention of pathogen infection of endangered species [57,58]. In other regions of the world, studies carried out to estimate the exposure of wildlife to livestock and zoonotic pathogens include commonly large sample sizes per species and long-term approaches. These may provide a more accurate understanding of disease epidemiology, allowing the identification of the main drivers of pathogen–wildlife interactions [59,60]. However, serological studies of pathogens present in wild cervids in Latin America are scarce, with very few samples per species [61,62,63,64], so the findings of this study, where a high circulation of livestock agents in wild pudus was identified, represent a significant contribution to the state-of-the-art epidemiological knowledge and to estimating potential conservation threats derived from the anthropogenic impact on wild ruminants in the region. To the best of our knowledge, no previous exposure data have been published for *Chlamydia abortus* exposure in wildlife from Latin America and the Caribbean. The findings on *T. gondii*, *Leptospira interrogans*, and *Neospora caninum* are also novel for wild ruminants in Chile. As occurs with most pathogen seroprevalence studies in wildlife, none of the serological tests used in this study have been validated for use in pudu, so the results should be interpreted with caution [65,66,67], and more serological and/or molecular evidence is required to complement our findings. This study faced limitations due to the selective analysis of pathogens, dictated by the finite volume of samples available. Not all pathogens were tested in each sample, which may affect the comprehensiveness of our findings. Despite this, the data presented offer important preliminary insights, and we advocate for subsequent research with broader testing to validate and expand upon these results.

### 4.1. Wild Animals

There are only two previous reports of seroprevalence for infectious agents in wild cervids in Chile [63,68], although with small sample sizes per species (<30) that are lower than those of the present study. Therefore, for pathogens with a low true prevalence, it is difficult to detect at least one infected individual in a population, which may be influencing the results [69,70].

#### 4.1.1. *Leptospira interrogans*

Leptospirosis is a good example of a disease that can affect the health of domestic and wildlife species and it is also a zoonosis [71]. With a wide variety of serovars that infect different species widely reported worldwide [72], there are few antecedents in wild ruminants in Latin America and no previous studies in Chile. The overall seroprevalence of *Leptospira interrogans* in wild pudus in Chile was on the reported average (18%) for wild Artyodactyl species in Latin America [73]. In other studies, the most detected serovars in wild deer were L. Grippotyphosa and L. Pomona, which have small rodents and pigs as their main hosts, respectively [74,75,76,77]. However, in the present study, the serovar Hardjo was detected with a higher frequency than that observed in other cervids species in LAC countries: (i) 5.6% in white-tailed deer in Mexico [78]; (ii) 0% in pampas deer in Argentina [61]; (iii) 0% in gray brocket deer in Bolivia [62]; and (iv) in mash deer, sambar deer, and pampas deer in Brazil (0–11.9%) [64,79,80,81]. The exposure to this serovar is similar to the highest rate reported in other regions of the world such as in elk (*Cervus elaphus*), mule deer (*Odocoileus hemionus*), white-tailed deer, and moose (*Alces alces*) in North America (0–11%) [77,82,83,84,85,86,87], and in red deer, fallow deer (*Dama dama*), and roe deer (*Capreolus capreolus*) in Europe (0–10.5%) [74,88,89,90]. In Chile, the seroprevalence of leptospirosis in cattle, which are recognized as the maintenance host of the serovar Hardjo [91] and to be a potential source of infection for humans and other animal species, was recently reported on the dairy farms of Los Rios and Los Lagos districts (5.3%), with the most prevalent serovars being Hardjo and Pomona [92]. The most frequent origin of wild pudus in the present study was Los Lagos district. All the pudu samples coming from this district were seropositive for Hardjo and Pomona serovars of *L. interrogans*, which makes it probable that the source of infection for the pudus was cattle from small farms, because they have the highest prevalence of *L. interrogans*, associated with a shortage of vaccination prevention programs against this disease [92]. In farmed deer species in New Zealand, the serovar Pomona appears to produce clinical and probably subclinical disease, whereas serovar Hardjobovis appears to cause only subclinical disease [93]. Thus, it seems of great relevance to evaluate the pathogenicity and clinical impact of both serovars in pudus.

#### 4.1.2. *Pestivirus*

Ruminant pestiviruses, such as *Pestivirus* A (formerly known as Bovine Viral Diarrhea Virus 1, BVDV-1), *Pestivirus* B (Bovine Viral Diarrhea Virus 2, BVDV-2), *Pestivirus* H (Bovine Viral Diarrhea Virus 3, BVDV-3, or HoBi-like *Pestivirus*, HoBiPeV), and *Pestivirus* D (Border Disease Virus, BDV), are widely distributed worldwide, causing abortions, mucosal disease, diarrhea, and respiratory problems in cattle and sheep [94]. Furthermore, they also caused outbreaks with high mortality in Pyrenean chamois (*Rupicapra pyernaica*) in the Spanish Pyrenees [95]. Despite the high relevance of pestiviruses in the livestock industry, where they generate significant economic losses [96], which places them on the list of notifiable diseases of the World Organization of Animal Health, the knowledge about the role of cervids in the epidemiology of this disease [94] and the impact of these pathogens on their health is not clear [97]. The seroprevalence of antibodies against *Pestivirus* found in this study in wild pudus was higher than that reported in most studies in wild cervids in Europe, North America, and Australia [98,99,100]. Higher seroprevalences were reported in Spanish red deer (19.5% and 10.8%) [101,102] and mule deer (17.1%) in the USA [77], suggesting sympatric cattle grazing alongside cervid populations as a source for BVDV infection. However, other studies propose that pestiviruses could be enzootic to wild cervid populations and thus maintained independently of livestock [98,100,103]. There are few studies focusing on detecting antibodies against pestiviruses in wild cervids in other LAC countries; until now, all animals were seronegative [61,62,81].

The ELISA used in this study has not been validated in pudu and cross-reactivity between different ruminants’ *Pestivirus* can occur [97,104]. Therefore, being unable to isolate the virus, it cannot be confirmed if the detected antibodies were against BVDV 1 or BVDV-2, reported in cattle in southern Chile [105,106], or against pestiviruses adapted to Chilean cervids [97]. In wild huemuls in Aysen district, 11.1% of the samples were positive for anti-pestivirus antibodies, although with a much smaller number of samples [67]. The only previous report of *Pestivirus* infection in wild pudu was an animal rescued in the Bio Bio region, infected with a *Pestivirus* of the BVDV-1b genotype isolated from lesions, suggesting the ability of the virus to cause clinical disease in pudus, posing a potential threat to the health of the wild population of this species [107].

In Chile, there are no reports of the presence of Border Disease Virus, whose reservoir host is sheep [108], while the seroprevalence of BVDV in cattle in the Los Lagos district is 3.5% [109]. For that reason, we suggest that the latter could potentially be the source of infection for the positive pudus in this study. Our study confirms the susceptibility and high circulation of pestiviruses in wild populations of pudu in Chile. However, more information is needed to understand the role of pudu in the epidemiology of the virus, e.g., if they can shed BVDV and if they can maintain BVDV without contact with cattle [110], and the impact of BVDV infection on the health and conservation of wild pudus.

#### 4.1.3. *Toxoplasma gondii*

*Toxoplasma gondii* is a globally zoonotic protozoan apicomplexa that infects a wide variety of warm-blooded animals [111], causing abortions and neonatal mortality in humans and domestic/wild ruminants [112,113,114]. Cervids have been reported as sentinels of environmental contamination by *T. gondii* and a potential source of clinical toxoplasmosis infection in humans by transmitting it through the ingestion of uncooked or undercooked meat containing tissue cysts [115,116]. The seroprevalence of *T. gondii* in wild pudu was low compared to that reported globally in other cervid species in wild environments [117,118], and for other wildlife species [1,119] and humans [120] in the same region of this study in Chile. However, as it is a species whose meat is consumed by local communities, the risk of contagion is latent and molecular studies are recommended to confirm the presence of infective parasitic stages in the muscles and characterize the genotypes of *T gondii* in pudus, as well as develop education and dissemination campaigns about the risks of consuming uncooked meat of this species.

#### 4.1.4. *Chlamydia abortus*

*Chlamydia abortus* is a Gram-negative intracellular bacterium recognized as having the most common causative agent of abortion in small ruminants and zoonotic public health problems with serious consequences in pregnant women and immunocompromised individuals [121,122]. However, few data are available on the prevalence and relevance of *Ch. abortus* in wildlife hosts [23,123,124]. *Chlamydia abortus* was recently reported in livestock in Latin America and the Caribbean countries [125]. However, to the best of our knowledge, this is the first description of *Ch. abortus* in South American wildlife. Our results suggest that it is not a common pathogen in the wild pudu populations in Chile. Further research will be necessary to determine the current epidemiological situation of *Ch. abortus* in domestic small ruminants and in other wild ruminants in Chile.

### 4.2. Under-Human-Care Pudus

Unlike the scarcity of infectious disease reports in free-ranging wild cervids in Latin America, in captive populations, there are epidemiological and pathological reports that confirm the susceptibility of native species in the region to livestock and zoonotic pathogens [5,37,126,127,128]. In the present study, the same infectious agents that were detected in wild pudus were detected in captive pudus, except for *Neospora caninum*, which suggests that, in Chile, infectious diseases of wildlife under human care are the same as those of free-ranging individuals. Therefore, ex situ conservation programs can directly benefit from medical research on captive species [129].

#### 4.2.1. *Neospora caninum*

*Neospora caninum* is a protozoan parasite that can cause neosporosis, a major cause of abortions and neonatal mortality in cattle, neuromuscular and neurological disorders in dogs, and, in some wildlife species (mainly in captive deer, rhinos, and carnivores), presenting with a variable clinical picture [130]. Cervids are recognized among the most important wildlife reservoirs for this pathogen [131] and a recent review describes that infection seroprevalence in deer was higher in South America compared with other regions of the world [131]. In Chile, *N. caninum* was previously reported in domestic animals [132], but not in wildlife [36]. Our results describe the pudu as a new deer host species for *N. caninum* and the first report of these protozoa in under-human-care wild ruminant species in Chile. However, contrary to reports in the region and worldwide, where wild cervids present higher seroprevalences than under-human-care animals [132], no antibodies were found in wild Chilean pudus, probably due to the low number of samples analyzed. Based on the history of abortions in captive pudus in Chile [5], and the perinatal mortality and outbreak of abortions recently reported in captive deer in Argentina [133,134], further research will be necessary to evaluate the pathogenicity of *N. caninum* in the pudu and the possible association with reproductive losses in this species.

#### 4.2.2. *Toxoplasma gondii*

The exposure of captive pudus to *T. gondii* was significantly higher than that found in free-ranging animals, which is different from what has been reported in other cervid species worldwide [117]. Exposure to this protozoan had previously been described in captive populations of cervids from other countries in the region [81] and in other wild species in Chile [36], but not in native cervids. The seroprevalence in under-human-care pudus was lower than that reported (38.3%) for other captive cervid species in Chile [45]. Although there are some reports of reproductive pathologies caused by *T. gondii* in deer species [135,136], reports of fatal cases of toxoplasmosis in cervids are not common in zoos or hatcheries [137,138]. Therefore, despite the high seroprevalence found for this protozoan, it should not be considered a major health threat to pudus under human care.

#### 4.2.3. *Leptospira interrogans*

The seroprevalence of *L. interrogans* in captive pudus was lower than that detected in wild pudus, as well as lower than that previously reported [37] in pudus in zoos in Chile. Serovar Hardjo, which was the most common in wild pudus, is reported for the first time in this species. Infection with this serovar had the highest prevalence in farmed deer species in New Zealand and cervids are suggested to be maintenance hosts [93]. However, in Brazil, clinical leptospirosis was reported in a pampas deer (*Ozotoceros bezoarticus*) by four serotypes of *Leptospira interrogans*, including serovar Hardjo [139], so preventive measures against this pathogen should be maintained in the pudu population to prevent disease occurrence. Likewise, it is recommended to deepen pathological studies to determine if there is clinical susceptibility in this species to the serovars reported here, and it may represent a threat in free-ranging animals or animals under human care.

#### 4.2.4. *Pestivirus*

To the authors’ knowledge, there are no reports of *Pestivirus* infection in captive populations of wild Artiodactyla species in other countries of Latin America and the Caribbean. The seroprevalence of antibodies against *Pestivirus* detected in under-human-care pudus was lower than that found in free-ranging pudus and significantly lower than that previously reported for pudus from a zoo in Chile (100%) between 2010 and 2012 [5]. Then, our results confirm the hypothesis suggested by these authors, that this outbreak was caused by a *Pestivirus* introduced to the pudu population maintained in the zoo, and not by an enzootic virus of the captive cervid populations in Chile. In addition, the co-occurrence of abortions with clinical and pathological signs of infectious origin with this epizootic in this zoo suggests BVDV as the cause for these abortions. After the removal of a pudu persistently infected with the virus, there have been no abortive events with signs of an infectious cause in this zoo during the last 10 years. There is no evidence of many species of cervids clinically susceptible to natural infection by BVDV [140], so future pathological and molecular studies are necessary to confirm the probable pathogenicity and impact of *Pestivirus* in pudu and other Chilean endangered cervids such as huemul and taruka.

#### 4.2.5. *Chlamydia abortus*

The seroprevalence of *Ch. abortus* in under-human-care pudus was higher than in the wild and, combined with several seroconversion events in pudus monitored longitudinally in this study, it confirms the susceptibility of this cervid to this pathogen. There is no history of reports of exposure to *Ch. abortus* in captive deer, so to the best of the authors’ knowledge, the pudu is the first deer species reported with evidence of infection by *Ch. abortus* in captive populations. Clinical and pathological studies are needed to determine if there is disease due to these bacteria in pudus.

### 4.3. Other Pathogens

The non-evidence of exposure in wild and captive pudus for pathogens commonly reported in cervids in other regions of the world, such as Bluetongue Virus and Epizootic Hemorrhagic Disease Virus, is expected, because they are none reported in Chile. Further, no prior evidence of these viruses in the country has been reported, and only recently were mortality episodes caused by EHDV reported in cervids in Latin America in Brazil [127]. Likewise, it is also expected that no evidence of exposure will be detected for some livestock and zoonotic pathogens, such as *Brucella abortus* and *Mycobacterium bovis*, which have a very low prevalence in the country in their host reservoir, cattle, thanks to the governmental programs for control and eradication [36,68]. Our results for *M. bovis* are similar to those reported through molecular screening of feces and serology for free-ranging pudus and huemul, where they found no evidence of infection throughout their entire distribution [68]. No evidence of exposure to Bovine Herpesvirus 1 was found in pudus, which is similar to what was reported in huemul [63], but different from studies developed in Europe, where this agent has been reported frequently in serological studies in free-ranging and captive cervids [141]. However, no mortality events due to BovHV 1 have been reported in cervids. SARS-CoV-2 has been detected only in captive and wild populations of white-tailed deer in the US [19,142], but not in other deer species in other regions of the world [143], so studies in cervid species phylogenetically close to the white-tailed deer, such as the pudu, would allow us to better understand its epidemiology. Our results may be influenced by the few samples analyzed or the low sensitivity of the non-specific technique for this species or a combination of all these factors. Reports on susceptibility to Hepatitis E Virus infection in deer are abundant in Europe [144]. However, it was not possible to confirm if pudus are susceptible to this virus, because, like SARS-CoV-2, the number of samples analyzed was very low, so it is recommended to analyze a greater number of samples to confirm our results. Finally, there is a recent report of molecular detection of *Coxiella burnetii* in a free-ranging pudu [145]; however, it is likely that the prevalence is very low or that the sensitivity of the technique is not the same, so it was not possible to detect it in the wild or captive populations of the present study. In addition to this, the prevalence and distribution of this pathogen in Chile are very low and localized, so it is suggested to analyze a high number of wild pudu samples using serological and molecular techniques that confirm the reported findings.

Finally, additional considerations should be taken into account regarding the results of rescued pudus, because they can be biased by some infectious pathogens that can cause ill animals. Clinical leptospirosis in deer cause signs like dullness [93], and *Toxoplasma gondii* infection influences the host behavior, including decreases in motor performance, learning capacity, neophobia, and fear; all of these alterations increase the probability of being attacked by a dog or suffering vehicle collisions, two main causes of admission of pudus in Chilean rehabilitation centers. Recently, evidence of *T. gondii* infection has increased risk behavior towards culling in red deer, supporting its role as a facilitator of predation risk [146].

The findings of the present study confirm previous evidence from studies in captive pudus on the susceptibility of this species to livestock diseases such as *Pestivirus*, *Leptospira interrogans* (serovars hardjo and Pomona) and provide new evidence on susceptibility to other livestock pathogens such as *Neospora caninum* and *Chlamydia abortus*. These results represent the first finding in Latin America and the Caribbean of a wild ruminant species with evidence of pathogen pollution by anthropogenic causes.

It is recommended to deepen epidemiological studies to characterize the role of pudu in the transmission dynamics of these pathogens as well as to develop studies to determine if they are pathogenic for the pudu. Additional factors to evaluate in these new studies are whether there is an influence of climate change and/or invasive exotic species, such as wild boar, red deer, or fallow deer, that share habitat with pudus in some areas throughout their distribution. All these factors can have a direct impact on the presence and abundance of several pathogens in wildlife and livestock species [147]. It is also recommended to inform communities that consume pudu meat about the risks of toxoplasmosis. Finally, it is suggested as a health surveillance tool to carry out studies on under-human-care wild species in zoos and hatcheries within their country of distribution as sentinels to determine the susceptibility of native species to infectious agents about which there is no epidemiological information.

## 5. Conclusions

This study represents the first multipathogen serological evaluation in pudu. The noticeable seroprevalence of livestock diseases such as *Pestivirus* and *Leptospira* Hardjo in wild pudus confirms the contact and transmission of livestock diseases to wildlife in Chilean template forest. According to our results, pudus may have a role as a wild reservoir of *Leptospira interrogans* serovar Hardjo and *Pestivirus*, and perhaps also for *Ch. Abortus* and *Toxoplasma gondii*. More research will be necessary for SARS-CoV-2 and Hepatitis E Virus, since the number of samples analyzed was so low.

## Figures and Tables

**Figure 1 animals-14-00526-f001:**
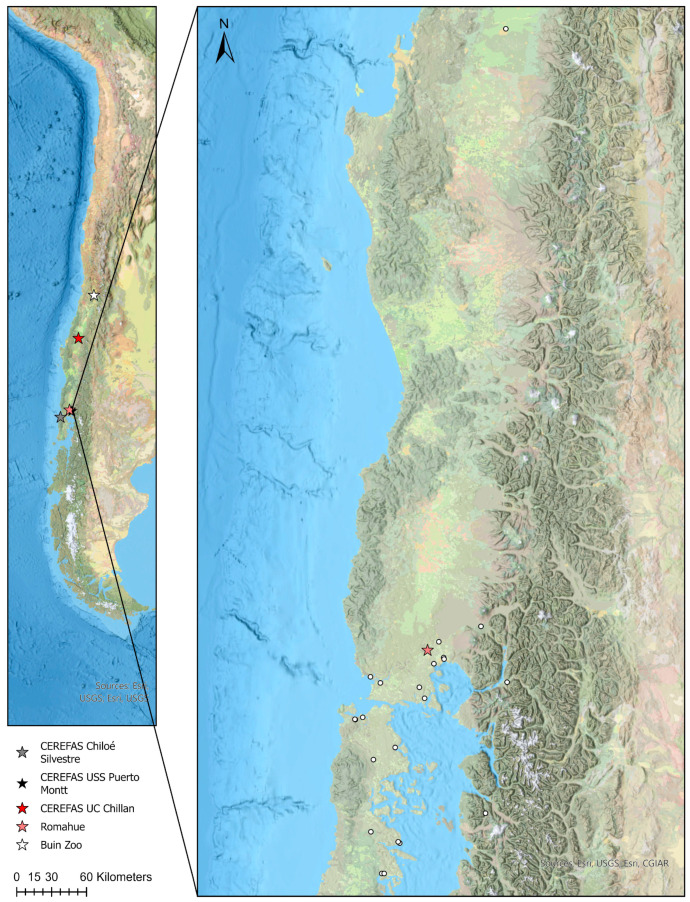
Map of pudus positives (white circles) for antibodies to pathogens surveyed in this research.

**Table 1 animals-14-00526-t001:** Serological test used in this survey.

Pathogen	Test	References
Bluetongue virus	INgezim BTV DR (Gold Standard Diagnostics, Madrid, Spain)	[27]
*Pestivirus*	VNT and INgezim *Pestivirus* Compac (Gold Standard Diagnostics, Madrid, Spain)	[5,6,7,8,9,10,11,12,13,14,15,16,17,18,19,20,21,22,23,24,25,26,27,28]
*Chlamydia abortus*	ID Screen^®^ Chlamydophila abortus Indirect Multi-species (IDvet, Grabels, France) and ELISA CHEKIT Chlamydophila abortus Antibody Test Kit, IDEXX Laboratories, Bern, Switzerland	[29]
*Coxiella burnetii*	PrioCHECK™ Ruminant Q Fever Ab Plate Kit (ThermoFisher Scientific, Waltham, MA, USA) and ELISA CHEKIT Q-Fever (*Coxiella burnetii*) Antibody Test Kit, IDEXX Laboratories, Bern, Switzerland	[30,31]
*Toxoplasma gondii*	ID Screen^®^ Toxoplasmosis Indirect Multi-species (IDvet, Grabels, France)	[32]
Epizootic Hemorrhagic Disease Virus	ID Screen^®^ EHDV Competition (IDvet, Grabels, France)	[33,34]
*Brucella abortus*	Rose Bengal test (Bengatestt, Parsippany, NJ, USA) and C ELISA (SVANOVIRt Brucella Antibody Test, SVANOVA, Uppsala, Sweden)	[35,36]
*Leptospira interrogans*	MAT (Pomona, Grippotyphosa, Copenhageni, Hardjo, Canicola)	[37,38]
*Mycobacterium bovis*	in-house P22 ELISA	[39]
Bovine Herpesvirus-1	VNT	[40]
SARS-CoV-2	VNT	[41]
Hepatitis E	ELISA	[42]
*Neospora caninum*	ELISA (CHEKIT Neospora caninum Antibody Test Kit, IDEXX Laboratories, Bern, Switzerland)	[43]

**Table 2 animals-14-00526-t002:** Seropositivity by *Pudu puda* condition (captive and free-range), 95% CI of the differences, and *p*-value for the comparison of seropositivity between groups.

Pathogen	*n*	Captive *	Free-Ranging *	Differences 95% CI		*p*-Value
				Lower	Upper	
*Pestivirus*	145	3/45 (6.67%)	8/100 (8.00%)	−0.0984	0.0889	0.384
*Leptospira* spp.	93	3/28 (10.71%)	10/65 (15.38%)	−0.2165	0.1231	0.787
*Toxoplasma gondii*	67	12/32 (37.50%)	3/35 (8.57%)	0.0677	0.5109	<0.001
*Neospora caninum*	32	2/21 (9.52%)	0/11 (0.00%)	−0.0996	0.2901	0.7731
*Chlamydia abortus*	72	5/39 (12.82%)	1/33 (3.03%)	−0.0502	0.2460	0.2847
Bluetongue virus	60	0/26 (0.00%)	0/34 (0.00%)		-	-
SARS-CoV-2	17	-	0/17 (0.00%)		-	-
Hepatitis E virus	20	-	0/20 (0.00%)		-	-
*Coxiella burneti*	74	0/35 (0.00%)	0/39 (0.00%)		-	-
*Brucella abortus*	73	0/31 (0.00%)	0/42 (0.00%)		-	-
BoHV-1	86	0/47 (0.00%)	0/39 (0.00%)		-	-
EHDV	60	0/26 (0.00%)	0/34 (0.00%)		-	-

* Values in brackets correspond to seropositivity expressed as percentage.

**Table 3 animals-14-00526-t003:** Final logistic regression model for risk factors for *Leptospira* spp. and *Toxoplasma gondii* positivity in *Pudu puda* individuals in under-human-care and free-range conditions from Chile.

Model	Variable	Categories	*p*-Value	OR	95% CI	
					Lower	Upper
*Leptospira* spp.	(Intercept)		0	0.138	0.066	0.289
	Age	Adult	Reference			
		Fawn	0.028	7.25	1.244	42.257
		Juvenile	0.649	0.604	0.069	5.29
		Indeterminate	0.176	7.25	0.412	127.7
*Toxoplasma gondii*	(Intercept)		0.213	0.632	0.307	1.301
	Condition	Under human care	Reference			
		Free-range	0.007	0.148	0.037	0.594

## Data Availability

The original contributions presented in the study are included in the article/Appendix A; further inquiries can be directed to the corresponding author/s.

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
