# Peer review of "High Exposure to Livestock Pathogens in Southern Pudu (Pudu puda) from Chile"

_animals, 2024, doi:10.3390/ani14040526_

Round 1

Reviewer 1 Report

Comments and Suggestions for Authors

Dear authors,

The manuscript “Monitoring Infectious Diseases in South America Deer Detects High Exposure of Southern Pudu to Livestock Pathogens” provides novel data on the presence of antibodies against several pathogens of animal and public health importance in pudus from Chile. This investigation has a significant merit since collecting samples from wild animals is not always easy. For those reasons, I think that these results must be published.

In contrast, I think that it is a very ambitious study, since it includes a large panel of pathogens, but not all samples were analysed for all pathogens; moreover, some pathogens were analysed in a very limited number of samples. Thus, some conclusions are not fully supported by the results. More detailed information is needed in the statistical analysis section since some results are surprising. In addition, the discussion section must be shortened. Therefore, major revisions are required before being accepted for publication.

FURTHER COMMENTS

-Title: Not very informative. Consider “High Exposure to Livestock Pathogens in Southern Pudu (Pudu puda) from Chile”

Abstract:

-Line 56: “was previously unknown”? It is known now, then?

-Line 56: Consider rewording: “presence of antibodies” or “seroprevalence”.

-Remove 95% CI in the abstract. They are not very informative. More results can be included after deleting these values.

-Lines 61-63: Do not include p values in the abstract. State that differences are significant.

-Line 66: Can you consider that prevalences of all pathogens are high?

Introduction:

-Line 75: It is

Materials and methods:

-How many samples were collected per animal? I guess than one per animal… but in line 169 it is stated that “UHC pudus sampled longitudinally”. This must be explained in detail in the materials and methods section. The 55 samples from UHC pudus are from different animals? How many UHC pudus were sampled? How many samples per UHC pudus were collected? The number of pudus sampled must be included in the materials and methods section (only in the results section).

-Line 113: Wild pudu sera originated from animals in rehabilitation centres. This can bias the study, since they can be ill animals with a higher probability of being predated, hunted or causing car crashes. This fact has to be further discussed.

-Line 141: How age was calculated?

-Line 142: several epidemiological data was gathered (Lines 125-126). Why only age, sex and condition was analysed?

-Lines 152-153: Include what R packages were employed in each analysis.

Results:

-Line 158: Why the number of analysis performed varied significantly between the included pathogens? What is the significance of having results of 17 animals for a single pathogen? In addition, no data of how many samples were analysed for each pathogen.

-Lines 159-167: This part would be easier to read and understand if all the results were summarized in a Table, including the number of samples analysed for each pathogen.

-Lines 167-168: Explain why some samples were not processed.

-Table 2: Change seropositivity expressed as proportion to percentage, easier to understand: i.e. 0.0667 to 6.7%. I am not sure what statistical analysis was performed, but results obtained by chi-square and Fisher tests for N. caninum and Chlamidia are not significant. Please explain.

-Line 188: Leptospira in italics. Also line 261-262. Review italics in the references section.

-Lines 190 and 192. Values are repeated in Tables 3 and 4. Please, delete Tables or delete the values in the text. Tables 3 and 4 can be merged.

Discussion:

-In general, discussion section must be shortened, especially some sections such as the Pestivirus section. I think that merging discussion of wild and UHC pudus could help to shorten this section and avoid repetitions.

-Lines 233-241: Are all those results comparable with those obtained in the present study? Perhaps it is better to state that that serovar can affect other wild cervids worldwide, but always showing low prevalences.

-Lines 245-246: Maybe a map could be interesting for representing if positives are present in a particular region. Different shape for wild and UHC purus and different colors for positives to different pathogens…. It is only an idea.

-Lines 449-451: It is stated that “However, they do not seem to contribute to the epidemiology of the rest of the studied infectious diseases in Chile. More research will be necessary for SARS-CoV-2 and Hepatitis E Virus.” Be careful with some conclusions since the number of samples analysed was so low.

-References section. Honestly, I think that 176 references are too many for an article. Reduce the number of them.

Comments on the Quality of English Language

Minor editing of English language required

Author Response

Response to Reviewer 1

Dear authors,

The manuscript “Monitoring Infectious Diseases in South America Deer Detects High Exposure of Southern Pudu to Livestock Pathogens” provides novel data on the presence of antibodies against several pathogens of animal and public health importance in pudus from Chile. This investigation has a significant merit since collecting samples from wild animals is not always easy. For those reasons, I think that these results must be published.

In contrast, I think that it is a very ambitious study, since it includes a large panel of pathogens, but not all samples were analysed for all pathogens; moreover, some pathogens were analysed in a very limited number of samples. Thus, some conclusions are not fully supported by the results. More detailed information is needed in the statistical analysis section since some results are surprising. In addition, the discussion section must be shortened. Therefore, major revisions are required before being accepted for publication.

R: Partially accepted. We appreciate for pointing this out. Our study did not extend to testing all pathogens in every sample due, because, as usually happens with serosurveys in wildlife, to limited sample quantities. Nonetheless, the results contribute valuable information, acknowledging the inherent challenges in sample collection. Between the lines 232-237, we have detailed this limitation within the manuscript to ensure the context of our findings is clear: “This study faced limitations due to the selective analysis of pathogens, dictated by the finite volume of samples available. Not all pathogens were tested in each sample, which may affect the comprehensiveness of our findings. Despite this, the data presented offer important preliminary insights, and we advocate for subsequent research with broader testing to validate and expand upon these results”,

FURTHER COMMENTS

-Title: Not very informative. Consider “High Exposure to Livestock Pathogens in Southern Pudu (Pudu puda) from Chile”

R: Thank by your recommendation. Corrected and included your title proposal.

Abstract:

-Line 56: “was previously unknown”? It is known now, then?

R: Accepted and changed in the lines 55-56: “A significant gap of exposure data for most livestock and zoonotic pathogens is common for several Latin America deer species”-

-Line 56: Consider rewording: “presence of antibodies” or “seroprevalence”.

R: Accepted and changed in the line 56: “This study examined the seroprevalence against…”

-Remove 95% CI in the abstract. They are not very informative. More results can be included after deleting these values.

R: We very much appreciate the comment. 95% CI was removed, and an additional result was incorporated in the abstract between the lines 63-65.

-Lines 61-63: Do not include p values in the abstract. State that differences are significant.

R: We very much appreciate the comment. All the p-values were removed from the abstract. 

-Line 66: Can you consider that prevalences of all pathogens are high?

R: No. We only consider as high, comparing with based evidence widely referenced in the discussion, the seroprevalence of pestivirus and Leptospira interrogans serovar Hadjo in pudu. The phrase in lines 66-67: “High seroprevalence of livestock pathogens such as Pestivirus and Leptospira Hardjo”

Introduction:

-Line 75: It is

 R: Accepted and changed in the line 76.

Materials and methods:

-How many samples were collected per animal? I guess than one per animal… but in line 169 it is stated that “UHC pudus sampled longitudinally”. This must be explained in detail in the materials and methods section. The 55 samples from UHC pudus are from different animals? How many UHC pudus were sampled? How many samples per UHC pudus were collected? The number of pudus sampled must be included in the materials and methods section (only in the results section).

R: We thank the reviewer's comment. A new text was included between the lines 162-167, and Tree new tables with this information were included (Table 2, 3, 4): “Although blood samples from 109 free-ranging (Table 2) and 55 captive (Table 3 and 4) pudu were collected, due to insufficient serum volume from some individuals, the sample size for specific serologic tests varied between 17 and 145 individuals, depending on the test. For all the the free-ranging pudus only one sample by pathogen was analyzed, however for captive pudus some individuals had one sample by pathogen (Table 3) and several others were monitored longitudinally (Table 4).”

-Line 113: Wild pudu sera originated from animals in rehabilitation centres. This can bias the study, since they can be ill animals with a higher probability of being predated, hunted or causing car crashes. This fact has to be further discussed.

R: We thank the reviewer's comment. However, we respectfully disagree. We reviewed evidence based knowledge about the infectious pathogens evaluated in this investigation and causes of admissions of deers to rehabilitation centers and did not find any evidence of your hypothesis. We appreciate if you had any evidence that support it for included this in the discussion.

-Line 141: How age was calculated?

R: All the pudu information was copied from the registered data of every zoo and rehabilitation center.

-Line 142: several epidemiological data was gathered (Lines 125-126). Why only age, sex and condition was analysed?

R: We very much appreciate the comment. Even when other variables were registered, this information was not registered for all the pudu that were part of the study, therefore these other factors (sample origin, sampling date, and health status, among others) were not considered in the risk factors analysis, given that this not registered values implies the removal of that pudu from the logistic regression.

-Lines 152-153: Include what R packages were employed in each analysis.

 R: We very much appreciate the comment. Information was added at lines 156-159.

Results:

-Line 158: Why the number of analysis performed varied significantly between the included pathogens? What is the significance of having results of 17 animals for a single pathogen? In addition, no data of how many samples were analysed for each pathogen.

R: This requirement is responded in the previous comments and the detail of samples by pathogen was incorporated in the new tables.

-Lines 159-167: This part would be easier to read and understand if all the results were summarized in a Table, including the number of samples analyzed for each pathogen.

R: We very much appreciate the comment. Sample size (n) was incorporated in table 5 to a better understanding of the results. The table is quoted in line 195. New data of sampled animals was included in the text between the lines 170-176: “All samples were negative for C. burnetii (39 samples from wild and 35 samples from captive), Brucella abortus (42 samples from wild and 31 samples from captive), M. bovis (24 samples from captive), Hepatitis E virus (HEV) (20 samples from wild), SARS-CoV-2 (17 samples from wild), Bovine Herpesvirus 1 (39 samples from wild and 47 samples from captive), EHDV (25 samples from wild and 25 samples from captive) and Bluetongue Virus (34 samples from wild and 26 samples from captive)”.

-Lines 167-168: Explain why some samples were not processed.

R: This requirement is responded in the previous comments.

-Table 2: Change seropositivity expressed as proportion to percentage, easier to understand: i.e. 0.0667 to 6.7%. I am not sure what statistical analysis was performed, but results obtained by chi-square and Fisher tests for N. caninum and Chlamidia are not significant. Please explain.

R: We very much appreciate the comment. Table 5 was corrected due to typo mistakes; percentages were incorporated, and n was added to each pathogen for better understanding of the results. The only pathogen with statistically significant differences was T. gondii (p < 0.001), N. caninum and C. abortus have p values > 0.05 to the test performed following the “Resource Selection” R package [44, 45 and 55 from the original manuscript]. Changes can be observed at line 195.

-Line 188: Leptospira in italics. Also line 261-262. Review italics in the references section.

 R: Accepted and changed in all the manuscript.

-Lines 190 and 192. Values are repeated in Tables 3 and 4. Please, delete Tables or delete the values in the text. Tables 3 and 4 can be merged.

R: We very much appreciate the comment. Numeric values have been removed from the text and tables 6 and 7 were merged. Changes can be observed at lines 208-211.

Discussion:

-In general, discussion section must be shortened, especially some sections such as the Pestivirus section. I think that merging discussion of wild and UHC pudus could help to shorten this section and avoid repetitions.

R: We very much appreciate the comment. However we respectfully disagree. The epidemiological conditions and available scientific information are totally different for wild and captive animals. Additionally 13 different pathogens and more 700 samples were analyzed for this research and we support that the current version of discussion is in agreement with the context.

-Lines 233-241: Are all those results comparable with those obtained in the present study? Perhaps it is better to state that that serovar can affect other wild cervids worldwide, but always showing low prevalences.

R: We thank the reviewer’s comment. However, we respectfully disagree. In this study, we analyzed 65 free-raging pudus for seven serovars of L. interrogans and found a high frequency of Livestock serovar Hadjo, we consider key to provide the reader with an updated introduction about the current knowledge regarding Leptoapira interrogans Hardjo in other deer species in the world.

-Lines 245-246: Maybe a map could be interesting for representing if positives are present in a particular region. Different shape for wild and UHC purus and different colors for positives to different pathogens…. It is only an idea.

R: A map with your recommendations was included.

-Lines 449-451: It is stated that “However, they do not seem to contribute to the epidemiology of the rest of the studied infectious diseases in Chile. More research will be necessary for SARS-CoV-2 and Hepatitis E Virus.” Be careful with some conclusions since the number of samples analyzed was so low.

R: Accepted and changed between the lines 465-471: “This study represents the first multipathogen serological evaluation in pudu. The high seroprevalence of livestock diseases such as Pestivirus and Leptospira Hardjo in wild pudus confirms the contact and transmission of livestock diseases to wildlife in Chilean template forest. According to our results, pudus could have a role as a wild reservoir of Leptospira interrogans serovar Hardjo and Pestivirus, and perhaps also for Ch. Abortus and Toxoplasma gondii. More research will be necessary for SARS-CoV-2 and Hepatitis E Virus since the number of samples analyzed was so low”.

-References section. Honestly, I think that 176 references are too many for an article. Reduce the number of them.

R: Thank you by your recommendation, the number of references is this version is 146.

Reviewer 2 Report

Comments and Suggestions for Authors

The paper addresses an interesting topic and deserves to be published, however, some issues must be addressed before reaching that stage:

1)      Avoid using unnecessary abbreviations like under human care (UHC), Latin America and the Caribbean (LAC) for example, that makes reading difficult.

2)      The number of samples used for the different determinations is not indicated.

3)      Although the authors mention having collected data on the samples and animals, they are not included in the article. It would be important to add this information relating these data to the results found

 4)      The program used to perform the statistical analyzes is not specified.

 5)      Finally, add that it is important to know the power of the test, that is, the probability of having accepted the correlation between the variables and that this is really true.

 Furthermore, given the great variation in the number of samples for each determination, the corresponding analysis should be carried out to know what number of data would be necessary for this value of r to be significant with a power of at least 0.8. The authors should add that statistical analysis.

Author Response

Response to Reviewer 2

The paper addresses an interesting topic and deserves to be published, however, some issues must be addressed before reaching that stage:

  • Avoid using unnecessary abbreviations like under human care (UHC), Latin America and the Caribbean (LAC) for example, that makes reading difficult.

R: Thank by your recommendation. Corrected in all the manuscript.

  • The number of samples used for the different determinations is not indicated.

R:  Thank by your recommendation. Three new tables (Tables  2, 3, 4) were included with detailed information about it and New data of sampled animals was included in the new tables and in the text between the lines 170-176: “All samples were negative for C. burnetii (39 samples from wild and 35 samples from captive), Brucella abortus (42 samples from wild and 31 samples from captive), M. bovis (24 samples from captive), Hepatitis E virus (HEV) (20 samples from wild), SARS-CoV-2 (17 samples from wild), Bovine Herpesvirus 1 (39 samples from wild and 47 samples from captive), EHDV (25 samples from wild and 25 samples from captive) and Bluetongue Virus (34 samples from wild and 26 samples from captive)”.

3)      Although the authors mention having collected data on the samples and animals, they are not included in the article. It would be important to add this information relating these data to the results found

R: Data of sampled animals was included in the new tables.

  • The program used to perform the statistical analyzes is not specified.

R: We very much appreciate the comment. Statistical software was mentioned at the end of the Material & Methods section under the Data analysis sub-section at lines XX - XX (“All the analyses were performed using R version 4.2.2 [49], and “fmsb” [50], “nlme” [51], “lme4” [52], “car” [53], “ggplot2” [54] packages for the multivariable logistic regression, and “Resource Selection” [55] package for the seroprevalence differences estimations.”), the software name is R, and the mentioned packeges were used to run all the statistical analysis of this study.

 5)      Finally, add that it is important to know the power of the test, that is, the probability of having accepted the correlation between the variables and that this is really true.

Furthermore, given the great variation in the number of samples for each determination, the corresponding analysis should be carried out to know what number of data would be necessary for this value of r to be significant with a power of at least 0.8. The authors should add that statistical analysis.

R: We very much appreciate the comment. As far as the authors' knowledge, the power of statistical analysis is run at the study design or usually done before the experiment is conducted, same with sample size. As this study is carried out by opportunistic sample size, we perform two tests that determine how the model adjusts to the data (Hosmer-Lemeshow Test), and an approximation to the R2 value from traditional regression models (McFadden pseudo-R2), both measures indicate how the model behaves in terms of adjustment to empiric data, and in terms of how much of the variability of the risk is explained by the factors in the model.

Round 2

Reviewer 1 Report

Comments and Suggestions for Authors

Dear authors

Although the reviewed version of the manuscript has been improved, there are some aspects that should be corrected before being accepted for publication

Line 58: definition of LZP is needed

Line 63: Replace “,” by “.”

Line 98: Remove “-“

Figure 1: Including a small representation of the whole country and the area enlarged (i.e. at the upper left corner) would be very useful for those readers who do not know the Chilean geography. I recommend including all free ranging animals in white dots (or positive white dots and negative black dots). I can only see one “under human care” triangle; represent the two populations, please.

Line 146: be careful if the “UHC” abbreviation was removed. Check it throughout the manuscript.

Lines 164-169: This is material and methods, since you are stating how many animals were sampled and problems for performing all analysis. In my opinion, Tables 2 to 4 are too large and I think they will be fantastic supplementary tables. In addition, Tables 3 and 4 can be merged.

Lines 174-175: be careful with this kind of typos

Lines 171-183: This information is included in Table 5. Do not repeat data in the text and in the table. For example: “Free ranging pudus presented antibodies against four pathogens (you can add them here altyhough it is not needed); the highest seropositivity rate was found for Leptospira and the lowest for C. abortus (Table X)”. The rest of the information is in the Table. Include the pathogens with negative results in the Table too.

Line 242: be careful when stating “pathogen positivity” since it could be interpreted as “active infection”. “seropositivity” could fit better.

Line 243 and 245: Table 6?

-Line 514: a prevalence of 8% is high? I suggest changing “high” by “noticeable”.

-Line 516: Change “could have a role” by “may have a role”.

-I had previously suggested that “Wild pudu sera originated from animals in rehabilitation centres. This can bias the study, since they can be ill animals with a higher probability of being predated, hunted or causing car crashes. This fact has to be further discussed.” There are extensive data on the interactions between predators and pathogens in animal populations. It has stated that some predators selectively remove infected hosts from a system because afflicted individuals generally exhibit pathogen-induced morbidity that increases vulnerability to predation (see Joly & Messier, 2004 for example). In addition, it is demonstrated that Toxoplasma gondii infection influences the host behaviour including decreases in motor performance, learning capacity, neophobia and fear, and increases in activity, and reaction times; obviously, infected animals have a higher probability of being predated, hunted or causing car crashes. In rodents, for example, behavioural changes (including spending more time in open, exposed areas, or be attracted by feline odour) increase their likelihood of consumption by a cat or any other predator (see for example a review by Johnson and Johnson, 2021). Although it is difficult to be extrapolated to animals, it was demonstrated that subjects with latent toxoplasmosis have significantly increased risk of traffic accidents than the noninfected subjects (Flegr et al, 2002).

Author Response

Dear author

Although the reviewed version of the manuscript has been improved, there are some aspects that should be corrected before being accepted for publication

Dear Reviewer

We appreciated all your comments; they have improved the quality of the manuscript.

Line 58: definition of LZP is needed

R: Accepted and changed: “Livestock and zoonotic pathogens antibodies….”

Line 63: Replace “,” by “.”

R: Accepted and changed: “than adults. In the case”.

Line 98: Remove “-“

R: Accepted and changed: “wildlife reservoir for zoonotic pathogens”

Figure 1: Including a small representation of the whole country and the area enlarged (i.e. at the upper left corner) would be very useful for those readers who do not know the Chilean geography. I recommend including all free ranging animals in white dots (or positive white dots and negative black dots). I can only see one “under human care” triangle; represent the two populations, please.

R: Accepted and changed.

Line 146: be careful if the “UHC” abbreviation was removed. Check it throughout the manuscript.

R: Accepted and reviewed all the manuscript.

Lines 164-169: This is material and methods, since you are stating how many animals were sampled and problems for performing all analysis. In my opinion, Tables 2 to 4 are too large and I think they will be fantastic supplementary tables. In addition, Tables 3 and 4 can be merged.

R: Accepted and changed.

Lines 174-175: be careful with this kind of typos

R: Accepted and fixed.

Lines 171-183: This information is included in Table 5. Do not repeat data in the text and in the table. For example: “Free ranging pudus presented antibodies against four pathogens (you can add them here altyhough it is not needed); the highest seropositivity rate was found for Leptospira and the lowest for C. abortus (Table X)”. The rest of the information is in the Table. Include the pathogens with negative results in the Table too.

R: Acepted and changed: “Free ranging pudus presented antibodies against four pathogens; the highest seropositivity rate was found for L. interrogans and the lowest for C. abortus (Table 2). Under human care pudus presented antibodies against five pathogens; the highest seropositivity rate was found for Toxoplasma gondii and the lowest for Pestivirus (Table 2).”

Line 242: be careful when stating “pathogen positivity” since it could be interpreted as “active infection”. “seropositivity” could fit better.

R: Accepted and changed: “between the tested factors and seropositivity only with”

Line 243 and 245: Table 6?

R: Accepted and changed.

-Line 514: a prevalence of 8% is high? I suggest changing “high” by “noticeable”.

R: Accepted and changed: “The noticeable seroprevalence of livestock diseases”

-Line 516: Change “could have a role” by “may have a role”.

R: Accepted and changed: “According to our results, pudus may have a role”.

-I had previously suggested that “Wild pudu sera originated from animals in rehabilitation centres. This can bias the study, since they can be ill animals with a higher probability of being predated, hunted or causing car crashes. This fact has to be further discussed.” There are extensive data on the interactions between predators and pathogens in animal populations. It has stated that some predators selectively remove infected hosts from a system because afflicted individuals generally exhibit pathogen-induced morbidity that increases vulnerability to predation (see Joly & Messier, 2004 for example). In addition, it is demonstrated that Toxoplasma gondii infection influences the host behaviour including decreases in motor performance, learning capacity, neophobia and fear, and increases in activity, and reaction times; obviously, infected animals have a higher probability of being predated, hunted or causing car crashes. In rodents, for example, behavioural changes (including spending more time in open, exposed areas, or be attracted by feline odour) increase their likelihood of consumption by a cat or any other predator (see for example a review by Johnson and Johnson, 2021). Although it is difficult to be extrapolated to animals, it was demonstrated that subjects with latent toxoplasmosis have significantly increased risk of traffic accidents than the noninfected subjects (Flegr et al, 2002).

R: Accepted, a new text was included in the manuscript between the lines 408-415: “Finally, additional considerations with the results of rescued pudues because they can be biased by some infectious pathogens that can cause ill animals. Clinical leptospirosis in deers cause signs like dullness [93]; and Toxoplasma gondii infection influences the host behaviour including decreases in motor performance, learning capacity, neophobia and fear. All these alterations increase the probability of being dog attacked or suffer vehicle collisions, two main causes of admission of pudus in Chilean rehabilitation centers. Recently, evidence of T. gondii infection increases risk behaviour towards culling in red deer, supporting its role as a facilitator of predation risk [146].”